# Application of Polymethylpentene, an Oxygen Permeable Thermoplastic, for Long-Term on-a-Chip Cell Culture and Organ-on-a-Chip Devices

**DOI:** 10.3390/mi14030532

**Published:** 2023-02-24

**Authors:** Linda Sønstevold, Maciej Czerkies, Enrique Escobedo-Cousin, Slawomir Blonski, Elizaveta Vereshchagina

**Affiliations:** 1SINTEF Digital, Department of Smart Sensors and Microsystems, Gaustadalléen 23C, 0373 Oslo, Norway; 2Institute of Fundamental Technological Research, Polish Academy of Sciences, Pawińskiego St. 5B, 02-106 Warsaw, Poland

**Keywords:** polymethylpentene (PMP), cell culture, oxygen control, microfluidic device, organ-on-a-chip

## Abstract

The applicability of a gas-permeable, thermoplastic material polymethylpentene (PMP) was investigated, experimentally and analytically, for organ-on-a-chip (OoC) and long-term on-a-chip cell cultivation applications. Using a sealed culture chamber device fitted with oxygen sensors, we tested and compared PMP to commonly used glass and polydimethylsiloxane (PDMS). We show that PMP and PDMS have comparable performance for oxygen supply during 4 days culture of epithelial (A549) cells with oxygen concentration stabilizing at 16%, compared with glass control where it decreases to 3%. For the first time, transmission light images of cells growing on PMP were obtained, demonstrating that the optical properties of PMP are suitable for non-fluorescent, live cell imaging. Following the combined transmission light imaging and calcein-AM staining, cell adherence, proliferation, morphology, and viability of A549 cells were shown to be similar on PMP and glass coated with poly-L-lysine. In contrast to PDMS, we demonstrate that a film of PMP as thin as 0.125 mm is compatible with high-resolution confocal microscopy due to its excellent optical properties and mechanical stiffness. PMP was also found to be fully compatible with device sterilization, cell fixation, cell permeabilization and fluorescent staining. We envision this material to extend the range of possible microfluidic applications beyond the current state-of-the-art, due to its beneficial physical properties and suitability for prototyping by different methods. The integrated device and measurement methodology demonstrated in this work are transferrable to other cell-based studies and life-sciences applications.

## 1. Introduction

Organ-on-a-chip (OoC) is a rapidly growing research field with the potential to replicate key aspects of human physiology in vitro and thereby serve as an innovative platform for precision medicine, drug screening, and studies of organ function and disease pathophysiology [1,2]. There is growing interest and need for novel materials and technologies that allow the retention of the native cellular environment in OoC and lab-on-a-chip platforms for long-term cell culture studies. Different combinations of material properties are required depending on the application, which is reflected in a wide range of diverse materials recently reported to be used for the fabrication of OoC devices. This includes elastomers (e.g., polydimethylsiloxane (PDMS), SU-8 polymers, polyester elastomers and thermoplastic elastomers), thermoplastics (e.g., poly(methyl methacrylate) (PMMA), polycarbonate (PC), cyclic olefin copolymer (COC), polystyrene (PS)), natural polymers (e.g., collagen, fibrin, alginate and hyaluronic acid), hybrid materials (e.g., PEG-fibrinogen and PLA-chitosan-gelatin), and adhesives and epoxy resins (e.g., medical grade glues, Norland Optical Adhesive 81 (NOA81) and off-stoichiometric thiol-ene-epoxy (ostemer)), 3D printing resins (e.g., UV-curable resins, dental resins and hydrogels) and inorganic materials (e.g., glass, silicon and ceramics) [2,3,4,5]. Many efforts have been put into the expansion of platforms’ functionalities via the integration of multiple materials [3,4,5] and sensing approaches [6,7]. However, often these platforms still compromise cellular experiments due to various material limitations and may prohibit direct comparison of results between different labs [8]. In this work, we demonstrate the novel application of the thermoplastic material polymethylpentene (PMP) for long-term on-a-chip cell culture, compare it to glass and PDMS, i.e., two commonly used materials for on-a-chip cell studies, and provide theoretical and experimental insights into oxygen supply and integration of oxygen monitoring in cell culture devices. 

### 1.1. Gas and Nutrient Supply to Cells in Culture

Continuous access to necessary nutrients and gases is a requisite when culturing cells, both in conventional cell culturing and when culturing cells in confined, enclosed spaces within OoC devices [2]. In conventional cell culturing, the culture medium is exchanged periodically to deliver nutrients and remove waste products produced by the cells. At the same time, secreted soluble factors are also removed. Gas supply is provided by diffusion from the surrounding air through the cell culture medium and is thereby continuous [9]. Constant O_2_ supply is necessary for all cells and cells cultured in media formulated with bicarbonate buffers also require 5% CO_2_ in the surrounding air to yield a physiological pH of 7.0–7.4 [2,10]. Lack of proper oxygen supply can have a major impact on experimental results through the activation of proteins from the hypoxia-inducible factor (HIF) family, which can rewrite cellular transcription programs in response to low-oxygen conditions [11,12]. In particular, a major role of HIF-1 has been described in regulating diverse cellular processes such as proliferation, apoptosis, glucose metabolism, proteolysis, etc. [11,12]. HIF-1 is currently drawing a lot of researchers’ attention for its role in cell survival in the low-oxygen microenvironments of tumors [12].

### 1.2. De-coupling of Gas Supply in Microfluidic Cell Culture Devices

For OoC and cell culture devices fabricated in gas-impermeable materials such as glass and common thermoplastics such as PMMA, PC, COC and PS [2], perfusion of the culture medium is the *only* source of supply of both nutrients *and* gas, as well as the removal of waste and secreted soluble factors [2,10]. However, media perfusion also exerts significant fluid shear stress on the cells, causing undesired cell-type specific and shear stress level specific responses [2]. Unless the effect of the shear stress itself is under investigation, it is beneficial to keep the perfusion flow rate, and thereby the shear stress, as low as possible [2]. Bunge et al. [10] compared important compounds which are either consumed or produced by cells (glucose, lactate, ammonium and oxygen) and found that oxygen has to be exchanged 10 times more frequently than the other compounds. Therefore, devices where the gas supply is *de-coupled* from the medium perfusion are highly beneficial. Furthermore, a constant medium exchange might also be highly undesirable from the perspective of the experimental aim itself, as it also removes from the cellular vicinity the unbound compounds that are under investigation (e.g., cytokines). Perfusion flow might also prove problematic when working with non-adherent cells, bacteria or viruses.

### 1.3. Use of PDMS Material in Cell Culture and Ooc Devices

De-coupling of the gas supply is one of the main reasons that the gas permeable elastomer material PDMS is so commonly used to fabricate microfluidic devices for cell culturing. Bunge et al. [10] compared microfluidic devices for long-term cultivation of mammalian cells and the gas was supplied by diffusion through PDMS in 13 devices out of 18 listed. The other devices relied on the perfusion of medium with dissolved gases, diffusion through hydrogel or open reservoirs. Wu et al. [13] reviewed microfluidic cell culture devices for the control of gaseous microenvironments in vitro, finding that out of the 36 devices listed, 34 were fabricated at least partially in PDMS. 

PDMS has some favourable properties as a research tool as it allows for rapid and cheap prototyping, facile bonding to glass, as well as favourable optical properties and deformability allowing integration of on-chip pumps and valves [14]. However, it also absorbs small hydrophobic molecules due to its porosity and hydrophobic nature, which affects the concentration of important solutes such as hydrophobic drugs and hormones [15]. Its high water permeability causes evaporation of water from PDMS microfluidic circuits and may lead to bubble formation and osmolality changes [14,16]. Other properties of PDMS that represent challenges for cell-based studies include the leaching of un-crosslinked oligomers [15], hydrophobic recovery after plasma hydrophilization which limits shelf life [17] and high compliance causing deformation of microchannels [14,18]. Further, it has become evident that large-scale production and commercialization of PDMS-based devices are hindered by up-scaling difficulties and low fabrication throughput [19]. Therefore, there is a need to identify new materials for the fabrication of gas-permeable OoC devices.

### 1.4. PMP Material for Long-Term Cell Culture Studies

As a gas-permeable, thermoplastic material used in extracorporeal membrane oxygenation systems [20], the material PMP may be a promising alternative, enabling oxygen control in devices intended for long-term cell studies. PMP, also known as TPX^TM^ (Transparent Polymer X) which is a trademark of Mitsui Chemicals, Inc., is a 4-methylpentene-1-based olefin copolymer with a characteristic molecular structure [21]. The low packing density of molecules causes very high gas permeation [21], making PMP very different from other thermoplastics which are nearly gas impermeable [2,14]. However, to date, PMP has scarcely been explored in microfluidic devices for cell cultivation and OoC.

Knoepp et al. [22] created a PMP-based microfluidic system for simultaneous Raman spectroscopy, patch-clamp electrophysiology, and live cell imaging to study key cellular events of single cells in response to acute hypoxia. The device was milled in PMP and attached to a commercial cell culture dish with a lowered glass bottom. The choice of PMP in this work was beneficial because it showed little interference with the Raman spectra of the target biomarkers. Ochs et al. [23] developed a computational model to predict oxygen levels in microfluidic devices of known oxygen permeability and cell types with known oxygen uptake behavior. The model was experimentally determined using microfluidic structures either injection molded in COC or PMP, or molded in PDMS, which were attached to commercial oxygen sensor foils. Oxygen levels inside the devices were monitored for two hours while adherent cells were growing on the sensor foils. Although these studies show promising results, much is yet to be investigated regarding the use of PMP for long-term cell culture devices, such as, e.g., oxygen monitoring during several days of cell culture, and culturing and imaging of adherent cells on the PMP material itself inside the devices. In the previous works of Knoepp et al. [22] and Ochs et al. [23] the cell-adherent part of the devices were fabricated from other materials.

Here, we present an investigation of the performance of the PMP material for devices for long-term on-a-chip cell cultivation and OoC applications. The assessment includes investigation of cell adherence, proliferation and morphology compared to glass, suitability for transmission light and fluorescence confocal microscopy of cells, compatibility with common chemicals, device fabrication and gas permeability for oxygen supply during 4 days of cell culturing in a sealed device in the absence of perfusion. To assess the gas permeability performance of PMP in such a setting, we developed a device with isolated, enclosed culture chambers fitted with oxygen sensors to allow the direct comparison between PMP and the control materials glass and PDMS (Figure 1). The enclosed, oxygen-monitored culture chambers serve as modules which may be used as building blocks in the formation of complex OoC and even body-on-a-chip devices with controlled oxygen environments by combining culture chambers, a suitable microfluidic channel network and chip-to-world interfaces in a singular design. 

Both prototyping methods (i.e., milling) [22] and technologies suitable for mass fabrication such as injection moulding [23,24,25] are applicable to PMP. This is a great advantage for reproducibility in medical and pharmaceutical research and commercial utilization of this material. The intention of this work is to demonstrate the potential of PMP for long-term microfluidic cell culture devices, and with this, provide an alternative for the academic and industrial communities interested in thermoplastic-based, oxygen-permeable cell culture devices.

## 2. Materials and Methods

### 2.1. Cells and Culture Conditions

A549 cells were originally purchased from American Type Culture Collection (ATCC, Manassas, VA, USA, cat. CCL-185). They were cultured in F12K medium (Thermo Fisher Scientific, Waltham, MA, USA, cat. 21127030) supplemented with 10% fetal bovine serum (Thermo Fisher Scientific, Waltham, MA, USA, cat. A3160802) and penicillin/streptomycin antibiotic solution (Thermo Fisher Scientific, Waltham, MA, USA, cat. 15140122). Cells were cultured in standard conditions (37 °C, 5% CO_2_) in a ThermoFisher series 8000 WJ incubator and passaged every 2–3 days upon reaching 90% confluence. 

### 2.2. Design and Fabrication of the Culture Chamber Devices

#### 2.2.1. Design of Culture Chamber Devices with Integrated Oxygen Sensors

The culture chambers were designed to test the applicability of PMP without compromising the experimental results by unnecessary design complexity. The devices as illustrated in Figure 2 were utilized for oxygen measurements (Section 2.3) and Hif1α staining (Section 2.4). Each device consists of a baseplate (second layer from the top in Figure 2) housing the cell culture chambers and a lid (top layer), both milled in PC. Each baseplate has two chambers for technical replicates in the experiments. Three different device variations with different materials integrated beneath the baseplate were designed to obtain adequate controls for the experiments with PMP: (1) device with PMP film (“sensor/PMP”) (Figure 2a); (2) device with a glass coverslip (“sensor/glass”) (Figure 2b); and (3) device with PDMS film (“sensor/PDMS”) (Figure 2c). The chamber diameter was 6.4 mm, similar to the diameter of wells in regular 96-well plates [26], and the height was 4 mm. The lids were designed to follow the contours of the baseplate, and a circular cavity (ID 10 mm, OD 13.6 mm, depth 1.2 mm) to insert 10 × 1.5 mm^2^ silicone O-rings (Otto Olsen, Skedsmokorset, Norway) ensured a tight seal between the baseplate and lid when secured by screws. Oxygen-sensitive spots SP-PSt3-NAU-D3-YOP with diameter 3 mm (PreSens, Regensburg, Germany, were glued to the lids using silicone glue Dowsil SG2 734 (PreSens, Regensburg, Germany) positioning them at the top centre of the chambers in the assembled devices. Circular sockets positioned 1 mm above the oxygen sensor spots were designed to accommodate an optical fibre to read the signal from the sensitive spots. The three materials compared in this study—PMP, glass or PDMS—were integrated at the bottom of the cell culture chambers. PMP film and glass coverslips were attached using pressure-sensitive adhesive (PSA) 92712 (Adhesives Research, ARcare^®^, Limerick, Ireland). For PDMS, a supporting clamping plate milled in PC was utilized in addition to PSA as illustrated in Figure 2c. The bond strength between PSA and PDMS is not sufficient and, therefore, additional mechanical clamping was required.

#### 2.2.2. Design of Culture Chamber Devices without Oxygen Sensors

For comparison of cell adherence, proliferation and viability (Section 2.5) on PMP film versus glass coverslips, the devices shown in Figure 3 were utilized. The only difference from the devices described in Section 2.2.1 is the lids, which are here only as support covers to ensure sterility. The devices will be referred to as “non-sensor/PMP” and “non-sensor/glass”.

#### 2.2.3. Fabrication

The baseplates, lids and clamping plates were milled in polycarbonate (Lexan) using a 3-axis milling machine (DMG DMC 1035 V, 10,000 rpm, rough/fine milling performed using 3/0.5 mm endmill, 500/300 mm/min feed rate, 3/0.05 mm axial depth of cut and 0.8/0.2 mm radial depth of cut). The milled parts were wiped with isopropanol, rinsed in DI water and ultrasonicated in DI water for 5 min. The oxygen sensitive spots were then glued to the lids and left to dry for 24 h at room temperature. Then, 18 sensors were characterized in air, finding a mean oxygen concentration of 21.5% with standard deviation of 0.03%. Commercial PMP film ME31110 (Goodfellow, Huntingdon, UK) with thickness 0.125 mm and grade DX845 was made optically transparent by polishing with diamond polish (Struers, Cleveland, USA, DP-suspension M (3 µm), DP-suspension P (1 µm), DP-lubricant blue) first with grain size of 3 µm, then 1 µm, and ultrasonically rinsed in DI water. Glass coverslips (Paul Marienfeld GmbH & Co, Lauda-Königshofen, Germany, #0101060) (thickness no. 1; 0.13–0.16 mm) or PMP film were attached to the baseplates with PSA, and PDMS film (Wacker, Munich, Germany, ELASTOSIL^®^ Film 2030 250/200) (thickness 200 µm) was attached using PSA and the clamping plate.

#### 2.2.4. Test of Compatibility with Different Sterilization Methods

PMP film was exposed to the following sterilization chemicals: Aerodesin 2000 (Medilab, Warsaw, Poland), 70% ethanol and 100% methanol. For physical sterilization, PMP film was exposed to either steam sterilization in an autoclave at 121 °C for 20 min (HICLAVE HG-113), UV light for 30 min in the laminar flow cabinet (MSC-Advantage; Thermo Fisher Scientific, Waltham, MA, USA) or plasma treatment. Plasma treatment was performed using a low-pressure plasma cleaner equipped with a 13.56 MHz / 50 W generator (Zepto, Diener electronic GmbH, Ebhausen, Germany) following the steps: (1) air in the working chamber was evacuated, and pressure was pumped down to 0.1 mbar, (2) oxygen was supplied to the working chamber (5 min, flowrate 10 sccm), and (3) the plasma treatment process was run (50 W, 5 min), maintaining oxygen flowrate 10 sccm.

### 2.3. Oxygen Measurements

Assembled devices were sterilized as follows: the base and the lid were wiped with Aerodesin 2000 solution. The base part was additionally exposed to UV light for 30 min. Both parts were washed with sterile PBS (Thermo Fisher Scientific, Waltham, MA, USA, cat. 20012-027) and air dried in the laminar hood. Culture chambers were coated with 0.01% solution of poly-L-lysine (Sigma-Aldrich, Burlington, MA, USA cat. P4707) for 10 min, then washed with PBS. Cells were detached from culture flasks, counted using the TC20 Automated Cell Counter (Bio-Rad Laboratories, Hercules, CA, USA), and seeded at the density of 25,000 cells per culture chamber in the full culture medium. Devices were covered with the non-sensor lids and cells were allowed to adhere overnight at normal culture conditions. At the start of the experiment, the medium was aspirated from the cells and 250 µL of fresh medium, pre-heated to 37 °C, was added to ensure identical initial conditions across all experimental variants. Immediately after medium exchange, the lids with sensor spots were installed onto the bases and tightly secured. The devices were put into the incubator and allowed to equilibrate to 37 °C before the first measurement. Devices were kept suspended 1 cm above the incubator shelf, to ensure proper airflow beneath them. For oxygen measurements, the devices were taken out from the incubator one at a time and put on the hot plate set for 37 °C (Figure 4a), to prevent them from cooling. Measurements were performed using an Oxy-1 SMA oxygen meter (PreSens, Regensburg, Germany) fitted with an optical fibre (PreSens, Regensburg, Germany, POF-L2.5-2SMA) and the data were collected with PreSens Measurement Studio 2 software. Five measurements were taken for each culture chamber per time point, with the following settings applied: temperature—37 °C; pressure—1025 hPa; mode—humid; and salinity—10 pmil. For each culture chamber and time point, the median of the five oxygen measurements was extracted. The average of the 2 technical replicates per biological replicate was found and used to calculate the average and standard deviation of the oxygen measurements for all biological replicates. The number of biological replicates was *n* = 3–5. 

### 2.4. Hif1α Staining

After completion of oxygen measurements as described in Section 2.3, the devices were opened, and the medium was aspirated. The cells were immediately washed with PBS buffer and fixed with 4% formaldehyde ( Sigma-Aldrich, Burlington, MA, USA, cat. 252549) in PBS for 15 min at room temperature. After further washing, cells were permeabilized with 0.1% solution of Triton X-100 ( Sigma-Aldrich, Burlington, MA, USA, cat. X-100) in PBS for 5 min to allow for staining of intracellular proteins, and then washed again. All washes were performed five times using PBS. Cells were blocked against unspecific binding with 5% Bovine Serum Albumin ( Sigma-Aldrich, Burlington, MA, USA, cat. A7906) in PBS for 1 h. Subsequently, cells were stained overnight at 4 °C with an anti-HIF-1 alpha antibody (Abcam, cat. ab179483) at a 1:500 dilution in the blocking buffer. After washing, secondary antibody (anti-rabbit-Alexa 488 IgG, Thermo Fisher Scientific, Waltham, MA, USA, cat. A21206) at a 1:1000 dilution was added for 1 h, at room temperature. Finally, cells’ nuclei were stained with DAPI ( Sigma-Aldrich, Burlington, MA, USA, cat. D9542) for 15 min and then the cells were washed again. Imaging was performed with Leica SP5 confocal microscope (Leica Camera, Wetzlar, Germany) with Leica LAS AF software, using HC PL APO 20x/0.70 objective. At least five fields of view were captured per culture chamber.

### 2.5. Live Cell Imaging and Calcein-AM Assay

Non-sensor devices with cells growing on PMP or glass bottom surfaces were periodically removed from the incubator and transferred to the Leica SP5 confocal microscope equipped with environmental chamber, allowing for maintaining same culture conditions (37 °C and 5% CO2) (Figure 4b). Transmitted light images of living cells were taken using Differential Interference Contrast (DIC). At least three fields of view were captured per culture chamber. At the end of the experiment, lids were removed, and the medium was aspirated. Calcein-AM ( Sigma-Aldrich, Burlington, MA, USA, cat. 56496) was diluted 1:2000 in a serum-free F12K medium, added to the cells for 20 min and then aspirated. After washing three times with medium, cells were immediately analyzed for fluorescence, indicating cell viability, under Leica SP5 microscope. All images were captured using HC PL APO 20x/0.70 objective and Leica LAS AF software.

### 2.6. O_2_ and CO_2_ Permeability of Polymers

The O_2_ and CO_2_ permeability of PMP film, PDMS film and polycarbonate was measured using the constant-pressure method, analogue to ASTM D3985—17 though employing gas chromatography analysis on the permeate stream, similarly as reported in [27], except the temperature was set to 37 °C. A detailed description of the procedure is found in the Appendix A.

## 3. Results and Discussion

### 3.1. Comparison of Cell Morphology and Proliferation on PMP and Glass

Figure 5 compares cell cultivation on standard glass coverslips and polished PMP film, both coated with poly-L-lysine prior to cell seeding. Images from the first day show a similar number of cells adhering to the both surfaces and time point for onset of cell adherence. The cell proliferation rate as judged by confluency development is also comparable, as is also the cell morphology on both materials. The calcein-AM staining after 48 h verifies that the cells are alive in similar numbers at the end of the experiment. 

As cell culture plates made from polystyrene [14] and glass coverslips are the most common materials used for cell cultivation and imaging [28], it is important to benchmark possible new materials against these. As for both polystyrene well plates [14] and often glass coverslips [29], PMP also had to be surface treated to facilitate cell adherence. In initial studies we compared non-treated PMP, plasma-treated PMP, poly-L-lysine-treated PMP and plasma plus poly-L-lysine-treated PMP. On non-treated PMP, cells scarcely adhered, and cell morphology was rounded and contracted. This was also observed for vascular smooth muscle cells and mouse embryonic fibroblasts (NIH 3T3), respectively, by Slepička et al. [30] and Michaljaničová et al. [31]. On PMP with either of the three treatments, cell adherence, proliferation and morphology were similar to Figure 5.

Plasma treatment is commonly used to increase the hydrophilicity of polymers and thereby increase cell adhesion and proliferation [14,30,31]. It is also the most common method for the fabrication of commercial polystyrene cell culture plates for adherent cells [14] (often referred to as “tissue-culture treated plates” or “TCPS—tissue culture polystyrene” plates). Slepička et al. [30] and Michaljaničová et al. [31] studied variants of plasma treatment of PMP to increase cytocompatibility for use of the material in tissue engineering. Slepička et al. used Ar plasma at low power (3 and 8 W), while Michaljaničová et al. used Ar or O_2_/Ar plasma at high power (50, 100 and 200 W). Other parameters were also varied. For all the variants of plasma treatment they show that the water contact angle is significantly reduced compared to non-treated PMP with a contact angle of approximately 100° and that cells successfully adhere and proliferate on the plasma-treated PMP as opposed to the non-treated PMP. However, they also demonstrate that after plasma treatment, the contact angle gradually increases towards the non-treated value again during the following days. For the low-power plasma treatments performed by Slepička et al., the reversal back to higher contact angles is faster than for the higher-power plasma treatments performed by Michaljaničová et al. where some of the samples remained at a 60° contact angle for at least 9 days. Other factors than water contact angle might also affect cell adherence, and more research is necessary to understand the time-dependent interplay of all experimental parameters. For the use of PMP for OoC and long-term on-chip cell cultivation, a plasma treatment yielding PMP surfaces stable over years after the treatment, as the case for commercial TCPS plates today, would be highly beneficial. This would ease collaboration between engineers and biologists in the lab-on-a-chip community, make experiments more practical and allow for mass production of ready-to-use devices.

In this work, poly-L-lysine treatment was selected as the method to increase cytocompatibility of PMP due to better compatibility with the experimental workflow. 

### 3.2. Suitability for Transmission Light and Fluorescence Confocal Microscopy

In microfluidic devices for long-term cell cultivation and OoC applications, it is essential to be able to continuously monitor the cells’ state with microscopy. As fluorescence staining and imaging may be detrimental to cells over time due to phototoxicity [32] and only make parts of the cells visible, transmission light microscopy is a preferred cell monitoring method. Figure 5 demonstrates that the appearance of the cells when imaged on glass and on PMP are similar, and hence that the optical properties of PMP are suitable for the application. 

Commercially available PMP films (Goodfellow, Huntingdon, UK) are extruded with one gloss and one matt side (Appendix A). To the best of our knowledge, fully transparent PMP films are currently not available commercially. Although fluorescent imaging is feasible on these commercial films, the matt side, characterized by a higher roughness, interferes with transmission light imaging, prohibiting reliable cell observations. This is demonstrated in Appendix A. To obtain PMP films suitable for microscopy a polishing method for the matt side was applied. The polishing procedure can be further optimized to prevent occasional scratches and bending of the PMP film due to the applied mechanical force. 

This paper reports, for the first time, the transmission light images of living cells growing on PMP. Slepička et al. [30] and Michaljaničová et al. [31] present fluorescent images of cells grown on plasma-treated PMP which are fixed and fluorescently labelled, however, without accompanying transmission light images. Nishikawa et al. [33] and Danoy et al. [34] cultured primary rat hepatocytes and cryopreserved primary human hepatocytes, respectively, on collagen type I-P coated PMP for more accurate evaluation of hepatocyte metabolism and drug screening purposes. However, transmission light microscopy images were not demonstrated as presented herein. We are not aware of other works where cells have been cultured on PMP. As stated above, transmission light microscopy for cell monitoring during cell culture in microfluidic devices is crucial and we, therefore, consider the results presented here as an important step in the elucidation of the suitability of PMP for organ-on-chip and long-term cell cultivation devices.

To study cellular mechanisms and responses in detail, high-resolution imaging techniques such as confocal microscopy are necessary. A major challenge with common microfluidic cell culture devices fabricated in PDMS is that due to the high elasticity of the material, a device thickness in the millimetre range is necessary to obtain suitable rigidity for robust channel geometry. To perform high-resolution imaging a working distance of generally less than 0.3 mm is necessary [35], hence millimetre-thick PDMS devices are not practical. Most objective lenses are designed to work with glass coverslips of thickness 0.17 mm (coverslip #1.5) when the specimen is in direct contact with the coverslip [36]. In our work, we used the commercially available PMP film of thickness closest to this and as thin as 0.125 mm. The stiffness of PMP, as it is a thermoplastic material, was suitable to constitute a robust, flat channel wall even at this thickness, allowing confocal microscopy as illustrated in Figure 5. According to the manufacturer, Mitsui Chemicals, Inc., PMP is transparent like glass and has an excellent transmission rate for visible light (>93%; haze < 5%) [37]. Additionally, PMP shows better transmission than glass in the UV range [37]. In summary, PMP appears to be highly qualified for cell imaging in general, as well as for the more specialized high-resolution imaging techniques such as confocal microscopy.

### 3.3. Compatibility with Common Chemicals for Cell Cultivation and Sample Preparation for Microscopy

Another important characteristic for materials to be used in microfluidic devices for cell culturing is the compatibility with common chemicals used in cell experiments. Due to its stable C-C bonds, PMP shows excellent chemical resistance, particularly against acids, alkalis and alcohols [21]. PMP was exposed to 4% formaldehyde and 0.1% Triton X-100 for cell fixation and permeabilization, in addition to Aerodesin 2000, 70% ethanol and 100% methanol for sterilization. Aerodesin 2000 consists of 32.5 wt% propan-1-ol, 18 wt% ethanol and 0.1 wt% glutaral. We did not observe any visible deterioration of the PMP film after exposure to any of these chemicals or any changes in its transparency. The same applied for the physical sterilization methods tested, namely plasma treatment, UV light exposure and steam sterilization. In fact, PMP is a predestined material for applications requiring steam sterilization as the water absorption is very low and, therefore, a dimensional change caused by hydrolysis cannot be observed [37].

Figure 6 shows confocal images of cells which are fixed, permeabilized and fluorescently labelled with anti-HIF-1-alpha antibody and Alexa 488-labelled secondary antibody, in addition to DAPI nuclear stain. This demonstrates both the PMP material’s compatibility with the fixation and permeabilization chemicals and its suitability for fluorescent imaging. DAPI and Alexa 488 fluorescence have excitation/emission maximums at 358/461 nm and 488/520 nm, respectively; however, according to Mitsui Chemicals, Inc., the transparency of PMP should allow similar results to glass in the entire visible range and an even better transmission than glass in the UV range [37]. As may also be seen in the fluorescent images in Figure 5 and Figure 6, we did not observe unspecific binding of any of the fluorophores to the PMP film. Based on this we can conclude that PMP is suitable for long-term microfluidic cell culture devices in terms of its chemical compatibility. 

### 3.4. Fabrication of Devices 

In this work, the experiments required a device in which gas influx was limited by the gas-permeable bottom wall while the rest of the device was fabricated in a gas-impermeable material (PC). For this, a film of PMP was manually aligned and laminated with the milled PC baseplate using a PSA (Figure 2). This straightforward prototyping method resulted in well-functioning proof-of-concept devices. For many OoC applications, however, fabrication of the entire device in PMP will be required and hence, alternative fabrication and bonding methods will be necessary, e.g., milling and injection moulding.

### 3.5. Adequate Gas Permeability to Sustain Cell Cultivation in Sealed Device without Culture Medium Perfusion 

Table 1 shows the measured O_2_ permeabilities of PMP film, PDMS film and polycarbonate. As seen from the table, the O_2_ permeability of the PDMS film (732.7 barrer) was approximately 23.5 times higher than of the PMP film (31.2 barrer). Polycarbonate only showed an O_2_ permeability of 1 barrer, hence very little O_2_ would diffuse through the polycarbonate walls of the devices, and we should be adequately able to compare the O_2_ permeation through the three different bottom wall materials PMP, glass and PDMS. 

Figure 7 shows the results from monitoring the O_2_-concentration in the three different device versions of the culture chamber module during four days of cell cultivation of A549 cells. As expected, the O_2_-concentration steadily decreased in devices with glass coverslips reaching approximately 10% after 10 h, 4% after 24 h and stabilize at around 3% on days 3 and 4. On the contrary, both devices with PMP film and PDMS film stabilized with O_2_-concentration at approximately 16% for the duration of the experiment. This demonstrates that, as the cells consumed oxygen, the O_2_ flux through both the PMP and PDMS film was high enough to supply the necessary oxygen. In a cell cultivation incubator with a humidified atmosphere with 5% CO_2_ at 37 °C, the O_2_-concentration is theoretically 18.6% [9]. The fact that the O_2_-concentration stabilized at 16% and not 18.6% has not been further investigated as the same observation was made for both PMP and PDMS film, and is unlikely to be linked to the gas flux through the films themselves when the O_2_ permeability of PDMS was 23.5 times higher than of PMP. CO_2_ monitoring was not performed in this work. However, as seen from Table 1, CO_2_ permeability was higher than O_2_ permeability for all materials and therefore shortage of CO_2_ supply should not pose a problem in this experimental setup. 

During the experiment, the cell growth was monitored by microscopy at least twice a day. As the oxygen sensor spots covered the center of the chambers, only the cells in the periphery of the chambers could be monitored while the devices were sealed, but at the end of the experiment (on day four) the devices were opened for inspection of the whole cell culture area. The development of cell confluency was similar in all devices, indicating that the difference in O_2_-concentration in devices with glass versus PMP and PDMS was not caused by lower cell numbers in the PMP and PDMS devices.

The trends for O_2_-concentration in all three types of devices were clear and showed very little fluctuation with time. In Figure 7 the average and standard deviation of the exact measured values from the experiments are plotted (data are not normalized). Therefore, there is a relatively large standard deviation in the results, especially on the first day for the devices with glass coverslips. However, as seen from Appendix A where the raw data from each replicate of the experiment are presented, the relatively large standard deviation is not caused by time-dependent fluctuations in O_2_-concentration within the same chamber, but rather the differences between experimental replicates (variations in exact cell seeding numbers, cell passage, incubator conditions, sensor reading errors, etc.).

After four days of cell cultivation and monitoring of O_2_-concentration, all the devices were opened and the cells were fixed, permeabilized, stained for HIF-1-alpha and inspected (Figure 6). We observed activation of HIF-1-alpha in numerous cells grown in devices with glass coverslips, where oxygen concentration dropped to approximately 3% at the end of the experiment according to the O_2_-concentration measurements. This activation was not visible in cells grown in devices with PMP, where the final measured oxygen concentration reached 16%. This is consistent with results presented by Uchida et al. [38], where the level of HIF-1-alpha was shown to rise first at 3% O_2_ in A549 cells subjected to a range of decreasing oxygen concentrations. It was not possible to obtain confocal images of the cells in devices with PDMS film, as the clamping plate hindered the access of the microscope objective.

Although the PMP film provided an adequate gas supply in the current work, this does not necessarily mean that the same applies to all device geometries and cell types used for microfluidic cell culture and OoC. As explained in the Appendix A, adequate gas supply depends on the cells’ oxygen consumption rate (OCR), number of cells per area, the permeability of the PMP and the PMP thickness, in addition to the oxygen concentration outside the device. According to Wagner et al. [39], measured rates of oxygen utilization by mammalian cells in culture range from <1 to >350 amol cell^−1^ s^−1^, and depend on cell type, function and biological status. According to the same paper, A549 cells as utilized in this work have an OCR of 27 amol cell^−1^ s^−1^ [39]. The number of cells per area depends on cell size, growth characteristics and importantly whether the device is used for 2D or 3D culture. The permeability of oxygen in PMP might vary between different grades of PMP and may be affected by the polymer prototyping method. The device geometry, i.e., wall thickness and percentage of the device made from the gas permeable PMP, might also vary greatly in different applications. Therefore, more research is necessary to further investigate PMP’s applicability for different cell types, cell culture formats (2D versus 3D) and microfluidic device geometries. An example of the effect of different cell types is found in the work by Ochs et al. [23] where oxygen concentration in 2 mm thick devices of COC, PMP and PDMS bonded to a gas impermeable oxygen sensor foil was measured. For endothelial cells, the oxygen concentration was maintained at 15–16% for both PMP and PDMS devices, while in COC the oxygen concentration reduced to 12% within the 2 h monitored. For hepatocytes, only the PDMS device maintained 16% oxygen while the PMP and COC device reduced to 9% and 4%, respectively, in 2 h. 

To better understand the adequacy of gas supply through PMP for different cell types in the culture chamber module developed in this project, an analytical analysis was performed. Using the equations for the flux of oxygen through culture medium (1) and PMP material (2) presented in the Appendix A, we compared oxygen flux in conventional cell culture with oxygen flux in the developed culture chamber module. A detailed description of these calculations is found in Appendix A in the supplementary information, and the summary of input parameters and calculated flux is found in Table 2. The calculations show that the flux of oxygen through the PMP film is 10 times and 20 times higher than through 100 µL and 200 µL medium in regular 96-well plates, respectively. The, 100–200 µL is the standard volume range used in 96-well plates. This implies that any cell line with adequate oxygen supply when cultured in regular well plates with standard medium volumes in 2D should have more than adequate oxygen supply when cultured similarly in the developed culture chamber module. 

## 4. Conclusions

In this work, we assessed various properties of the PMP material and showed that the material is highly suitable for long-term cell culture and OoC devices. Most importantly, we have shown that PMP has comparable performance to PDMS in terms of oxygen supply in a sealed culture chamber with A549 cells cultured for 4 days. For both materials, the oxygen concentration stabilized at 16%, while with glass the oxygen concentration reduced to 3%. Analytical analysis showed that the theoretical oxygen flux through PMP film is 10 and 20 times higher than the flux through 100 and 200 µL medium in 96-well plates. Hence, any 2D cell line cultured in standard conditions should have more than adequate oxygen supply in the presented culture chamber module. Adequate gas supply through the device material itself enables de-coupling of the oxygen supply from the culture medium perfusion, which allows slower perfusion rates and less shear stress on the cells. This, ultimately, permits better control of the cell microenvironment in OoC devices. 

We have also shown that cell adherence, proliferation, morphology and viability of A549 cells are similar on PMP film and glass coverslips coated with poly-L-lysine. We presented the first-time transmission light images of cells growing on PMP, demonstrating that the optical properties of PMP are suitable for non-fluorescent, live cell imaging. As opposed to PDMS, PMP, due to a combination of favourable optical properties and stiffness, even for a film as thin as 0.125 µm, was very suitable for high-resolution confocal microscopy. In addition, we showed that PMP is compatible with physical and chemical device sterilization (steam sterilization, plasma treatment, UV light, ethanol, methanol and Aerodesin 2000), cell fixation (formaldehyde) and permeabilization (Triton X-100) and fluorescent staining. 

In summary, PMP holds many advantegous properties for microfluidic devices for long-term cell cultivation. With further development of device prototyping, we believe that PMP will play an important role in the field in the years to come.

## Figures and Tables

**Figure 1 micromachines-14-00532-f001:**
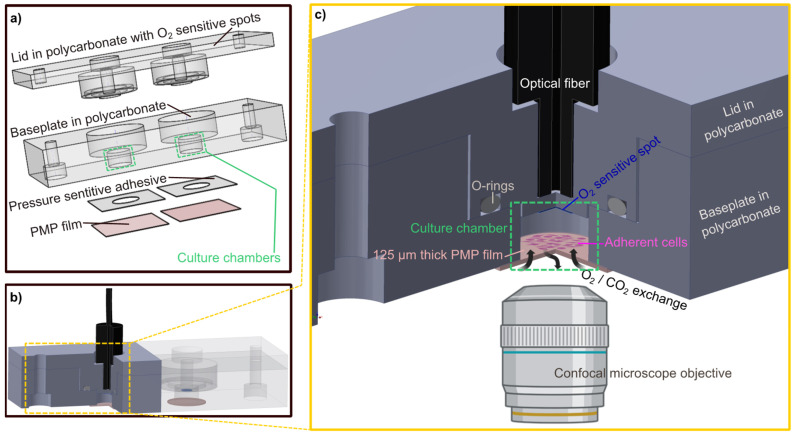
**The developed culture chamber device.** Culture chamber device designed to allow assessment of the PMP material. (**a**) An exploded-view drawing of the individual parts of the culture chamber device: the lid milled in polycarbonate with O_2_ sensitive spots attached, the baseplate milled in polycarbonate where the dimensions of the culture chambers are defined (green), a pressure sensitive adhesive (PSA) layer and the gas permeable PMP film. (**b**) The assembled culture chamber device with the optical fiber inserted into the customized cavity. For better visualization, the polycarbonate parts are made transparent in the right half of the device while they are filled with solid colors in the left half where a cross-sectional view exposes the culture chamber. O_2_-sensitive spots (blue) and PMP film (pink) are illustrated. (**c**) An enlarged schematic of the section marked yellow in (**b**). The cross-sectional view of the device exposing the culture chamber and its components. The 3D device images were created with Inventor 2022, illustrations of microscope objective and adherent cells were created with BioRender.com (accessed on 21 November 2022).

**Figure 2 micromachines-14-00532-f002:**
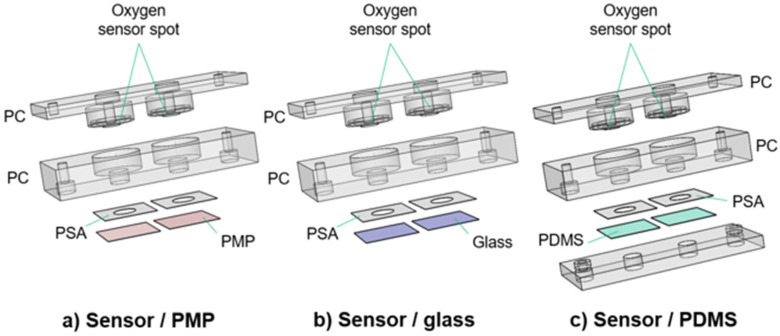
**Culture chamber devices with integrated oxygen sensors.** Three different designs of the culture chamber device. The main device is the sensor/PMP device, and the two remaining devices were necessary to obtain adequate controls for the experiments with PMP. Each device has in total four (**a**,**b**) or five (**c**) layers. From the top, the device layers are (1) the lid milled in PC (**a**–**c**); (2) the baseplate milled in PC (**a**–**c**); (3) PSA (**a**–**c**); (4) PMP film (**a**), glass coverslip (**b**) or PDMS film (**c**); and (5) the clamping plate (**c**). See Section 2.2.1 for details. These devices were utilized for oxygen measurements (Section 2.3) and Hif1IF-1α staining (Section 2.4).

**Figure 3 micromachines-14-00532-f003:**
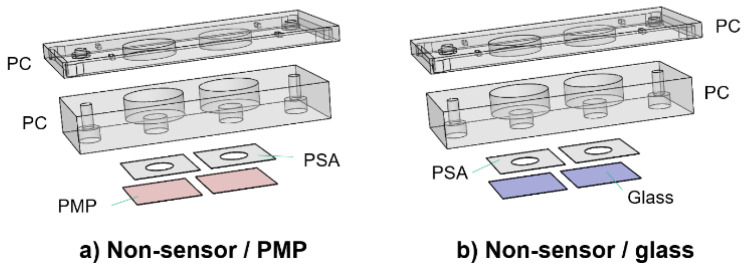
**Culture chamber devices without oxygen sensors.** Two devices used for the comparison of cell adherence, proliferation and viability (Section 2.5). The only difference from the devices in Figure 2 is in the design of the lids, which are here only support covers to ensure sterility (referred to as “non-sensor lids”). These were designed with either a PMP film (**a**) or a glass coverslip (**b**) for the bottom part of the culture chamber.

**Figure 4 micromachines-14-00532-f004:**
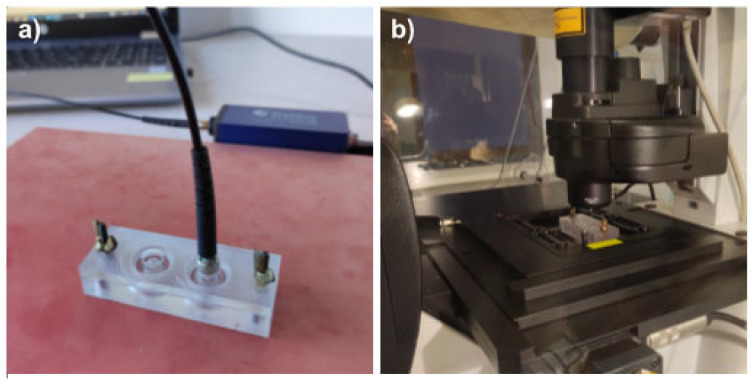
**Oxygen measurements and confocal microscopy.** (**a**) Demonstration of a cell culture device placed on the hot plate set to 37 °C while measuring the oxygen concentration by insertion of the optical fibre into the top cavity for readout of the signal from the fluorescent oxygen-sensitive spots integrated into the lid. (**b**) Demonstration of the cell culture device standing on the Leica SP5 confocal microscope equipped with environmental chamber set to 37 °C and 5% CO_2_.

**Figure 5 micromachines-14-00532-f005:**
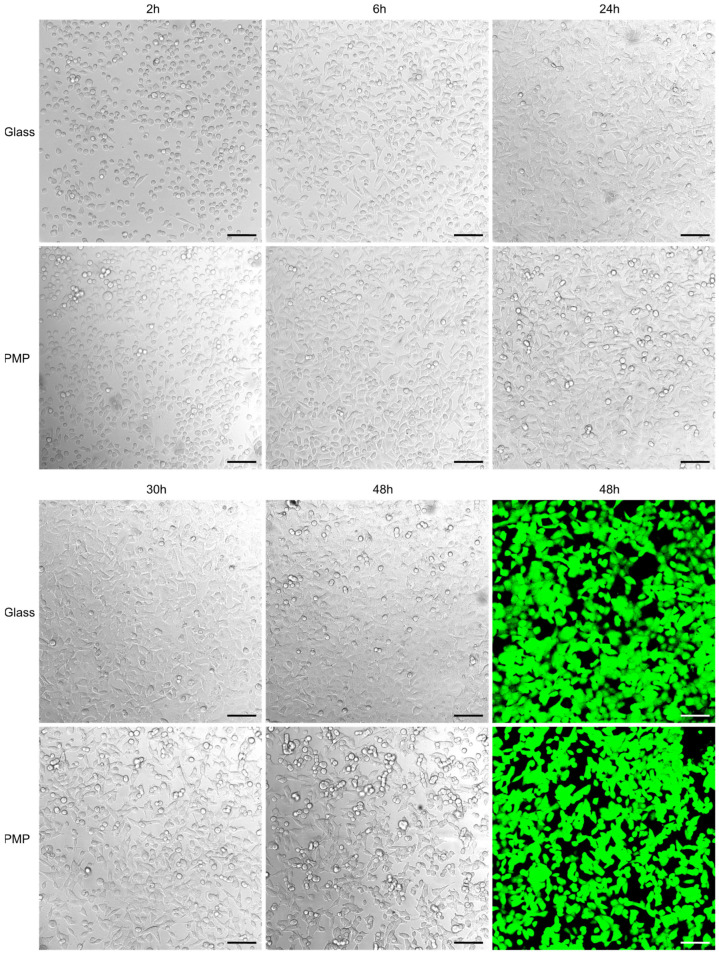
**Cell adherence, proliferation and viability** Growth and viability of A549 cells cultured inside the non-sensor/glass (“glass”) or non-sensor/PMP (“PMP”) devices (see Figure 3 and Section 2.2.2 for details on device designs). Both materials were coated with poly-L-lysine prior to seeding the same number of cells on both glass and PMP. Succeeding images show initially adhering cells (at 2 h), assuming of typical elongated morphology (6 h) and successful proliferation on both materials (up to 48 h) during culture at 37 °C and 5% CO_2._ The last images show the same cells stained with calcein-AM to verify their viability (green fluorescence indicates living cells). Scale bar: 100 µm.

**Figure 6 micromachines-14-00532-f006:**
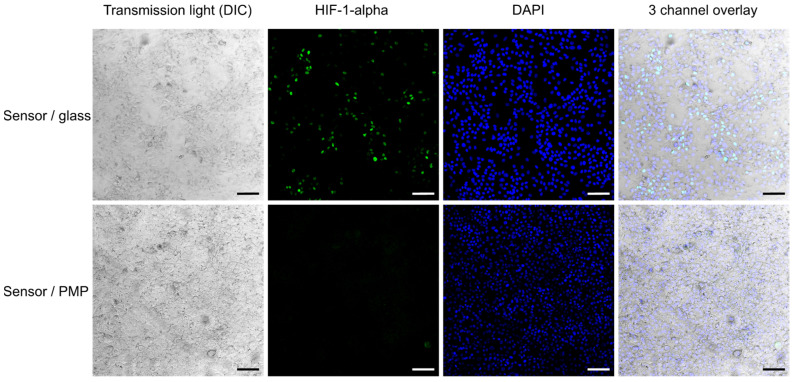
**HIF-1α activation.** Activation of the HIF-1α transcription factor in A549 cells cultured in sensor/glass and sensor/PMP devices (see Figure 2 and Section 2.2.1 for details on device designs) after the last measurement of oxygen concentration at 75 h after sealing of devices (see Figure 7). Cells were fixed with formaldehyde before fluorescent staining. The second column shows the presence of activated HIF-1α (green) in numerous cells cultured in the sensor/glass device, while almost none was observed in the sensor/PMP device. Transmitted light images and an overlay with DAPI nuclei staining (blue) show the morphology of a highly confluent cell monolayer. Scale bar: 100 µm.

**Figure 7 micromachines-14-00532-f007:**
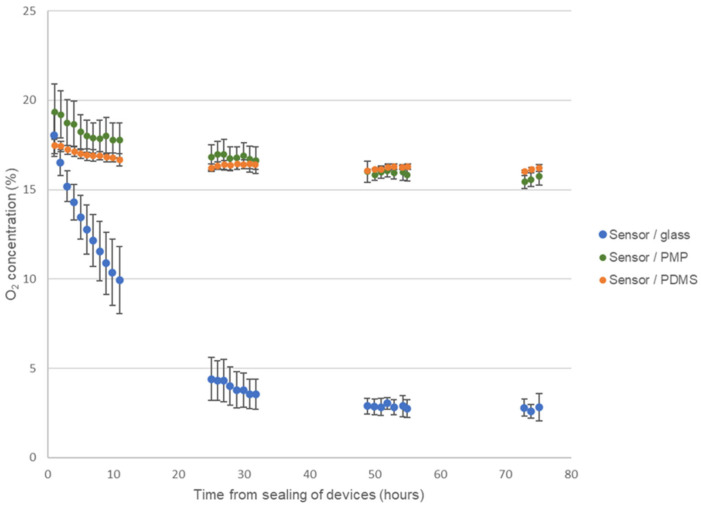
**Monitoring of oxygen concentration.** Results from monitoring the oxygen concentration during four days of culture of A549 cells in the three different device versions of the culture chamber module, sensor/glass, sensor/PMP and sensor/PDMS (see Figure 2 and Section 2.2.1 for details on device designs). Data are presented as average ± SD, *n* = 3–5.

**Table 1 micromachines-14-00532-t001:** Measured O_2_ and CO_2_ permeabilities.

Sample	O_2_ Permeability (Barrer)	CO_2_ Permeability (Barrer)
PDMS film	732.7	3246.4
PMP film	31.2	90.7
Polycarbonate	1.0	3.7

**Table 2 micromachines-14-00532-t002:** Overview of input parameters (light grey) and calculated oxygen flux (dark grey) through the PMP film compared to 100 µL medium and 200 µL medium in regular 96-well plates. See Appendix A for details.

Oxygen Flux Through:	P (Permeability) or D (Diffusion Coefficient)	Δp (Pressure) or ΔC (Concentration)	Δx (Thickness) (µm)	F (Flux) (pmol cm^−2^ s^−1^)
PMP film	31.2 barrer	141.4 mmHg	125	157.5
100 µL medium in 96-well plate	2.69 × 10^−5^ cm^2^/s	0.183 mM	3125	15.75
200 µL medium in 96-well plate	2.69 × 10^−5^ cm^2^/s	0.183 mM	6250	7.876

## Data Availability

Data is contained within the article and Appendix A.

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
