# Peer review of "Application of Polymethylpentene, an Oxygen Permeable Thermoplastic, for Long-Term on-a-Chip Cell Culture and Organ-on-a-Chip Devices"

_micromachines, 2023, doi:10.3390/mi14030532_

Round 1

Reviewer 1 Report

The work of Sonstevold et al., presents the ability of the material polymethylpentene (PMP) for long-term on-a-chip cell culture. Even if they provide results showing good properties of the material compared to glass or PDMS, I don't think the work can be accepted in the present form. The main reasons for the decision are the following:

The authors relate their chip as microfluidic, even if no measurements reporting flow or flow-simulations are described. 

Furthermore, they relate their results as "long-term" on-a-chip culture, even if their results of cell compatibility are reported for only 48h. 

The novelty of using PMP instead of PDMS or other polymers and materials used generally for OoC (such as dental resin) is not well described.

Minor comments:

- Figure 1 is not clear as well as the description of the chamber reported in lines 135-139. For the Figure: where is allocated the microfluidic chip? As it is now it means that in the chamber only a glass coverslip with adherent cells can be allocated in order to do the O2 measurements. Which are its dimensions? Regarding the sentences in lines 135:139: a better explanation of the reason for the chamber should be added. Furthermore, which is the relationship between the chamber and the possibility to develop complex Ooh or body-on-a-chip? Please better specify this part.

-  Figure 5: scale bar are not reported. Furthermore, the images in BF are in a very low resolution whereas the images in green are quite saturated. Please revise the Figure accordingly. 

-Figure 6 is also missing scale bar.

Reviewer 2 Report

In this paper, the authors demonstrate polymethylpentene films for cell culture and organ-on-a-chip microfluidic devices. However, I suggest addressing the following comments.

1. The Abstract and Introduction sections should be revised to be brief and concise. The subheadings in Introduction were not suggested to be used.

2. The novelty of this work should be addressed, especially the performance superiority compared with other types of materials or devices. The development in this field should be included in detail in the Introduction section.

3. The cell viability data could also be plotted to directly support and explain Figure 5.

4. Figure 7 should be recreated with standard journal format for better demonstration.

5. Why just cell tests of A549 were conducted? How about other typical types of cells, which can be added to further verify its superior performance.

6. References should be checked, such as formatting errors. Some recent advances in this field are suggested to be added.

Round 2

Reviewer 1 Report

Sønstevold et al., well replied to my previous comments. So I think now the study is ready for being published. 

Reviewer 2 Report

Good luck.